# Long-Term Outcomes of Endovascular Embolization in a Vein of Galen Aneurysmal Malformation: A Single-Center Experience

**DOI:** 10.3390/diagnostics13162704

**Published:** 2023-08-18

**Authors:** Chingiz Nurimanov, Yerbol Makhambetov, Karashash Menlibayeva, Nurtay Nurakay, Nursultan Makhambetov, Elena Zholdybayeva, Serik Akshulakov

**Affiliations:** 1Vascular and Functional Neurosurgery Department, National Center for Neurosurgery, Astana 010000, Kazakhstan; yermakh@gmail.com (Y.M.); nurtaynurakay92kz@gmail.com (N.N.); nmahambetov7@gmail.com (N.M.); raim@rambler.ru (S.A.); 2Hospital Management Department, National Center for Neurosurgery, Astana 010000, Kazakhstan; karmen@mail.kz; 3National Scientific Shared Laboratory, National Center for Biotechnology, Astana 010000, Kazakhstan; lenazhol@gmail.com

**Keywords:** AV malformation of the vein of Galen, the vein of Galen aneurysmal malformation, endovascular embolization, Bicêtre score, neonates, hydrocephalus

## Abstract

Background: A vein of Galen aneurysmal malformation (VGAM) is a rare congenital cerebral vascular condition with a high mortality rate if left untreated. This study describes the long-term outcomes of patients with VGAM, who were treated with endovascular embolization. Methods: This retrospective analysis focused on VGAM patients who underwent one or more endovascular embolization sessions between January 2008 and December 2022. The study included newborns and children under 18 years. Data encompassed clinical and demographic characteristics, types of endovascular embolization, treatment complications, mortality rates, and long-term outcomes. Results: Out of 22 VGAM cases, the majority were boys (86.36%), and the average age of the participants was 38 months, ranging from 25 days to 17 years. Endovascular embolization using liquid embolizing agents was the most common intervention (50%), and around 73% of patients underwent multiple sessions. Some patients underwent ventriculoperitoneal shunting (VPS) due to persistent hydrocephalus. In long-term outcomes, four patients (18.2%) showed developmental delays, and 16 patients (72.7%) had a positive outcome. Conclusions: Combining endovascular therapy with a comprehensive management strategy significantly reduces mortality rates and improves the possibility of normal neurological development in patients.

## 1. Introduction

A vein of Galen aneurysmal malformation (VGAM) is a rare congenital cerebral vascular condition that arises during fetal development between 6 and 11 weeks [1]. It accounts for approximately 1% of all congenital malformations and around 30% of vascular malformations in children, with an incidence of less than one in 25,000 cases [2]. VGAM presents with an abnormal connection between the choroidal arteries and the precursor of the vein of Galen, known as the median prosencephalic vein of Markowski (MPrV, MV) [3], which results in excessive blood flow in the venous system, affecting the pulmonary circulation and leading to various multi-organ complications.

The etiology of VGAM remains poorly understood, although some genetic and molecular factors have been found to be associated with vascular malformations [4]. Although specific genes associated with VGAM are unknown, some genes are linked to VGAM coexistence with other syndromes, such as p120-RasGAP (RASA1) mutations in cutaneous malformations-arteriovenous malformations (CM-AVM), as well as isolated mutations in activin A receptor type II-like 1 (ACVRL1) and endoglin (ENG) in hereditary hemorrhagic telangiectasia [5].

VGAM is identified in early life with signs of congestive cardiac failure in newborns. If left untreated, it leads to a high mortality rate [6], with only about 2.5% of cases experiencing spontaneous thrombosis [7]. Historically, open microsurgery was the most common treatment strategy for VGAM. In contemporary practice, endovascular embolization (EE) has emerged as the preferred and more frequently employed treatment option, due to its safety, effectiveness, and decrease in mortality rates, which is around 15.7%, while open surgery carries a higher mortality rate of 84.6% [8]. EE involves blocking the feeding arteries or draining veins to reduce excessive blood flow.

In the modern endovascular era, tailoring the approach to each patient’s unique circumstances becomes essential to ensure optimal outcomes. Given the intricate nature of VGAM and the challenges they pose, various techniques are often necessary within the endovascular approach to ensure both safety and effectiveness. These techniques encompass a range of flow-control methods and regionally targeted strategies [9]. It is crucial to factor in the procedural thoroughness when addressing VGAM, as incomplete embolization has been linked to unfavorable clinical outcomes [10]. This underscores the importance of a comprehensive and precise endovascular approach in managing VGAM cases.

Recent advancements in endovascular techniques have brought forth various approaches for treating VGAMs, including transarterial, transvenous, or combined methods. This progress has led to the development of novel tools for endovascular procedures, including microcatheters featuring detachable tips. Furthermore, innovative types of embolizing materials have been formulated [11], notably enhancing the efficiency and safety of EE, which have been documented in various meta-analyses. According to Yan et al.’s findings [12], EE led to good clinical results in approximately 68% of patients with VGAM, whereas mortality and complications were observed in 10% and 37% of cases, respectively. This viewpoint is supported by other reviews, which also highlight positive post-EE treatment clinical outcomes in over 60% of patients [2,10]. This positions EE as a more beneficial treatment choice for VGAM, and endovascular intervention continues to exhibit its superiority over conservative management or traditional surgical methods.

Being a rare disease, VGAM is of special interest to interventional neurosurgeons and radiologists as the endovascular approach stands as the only safe therapeutic option, making it even more compelling. Thereby, this study presents the long-term follow-up outcomes of EE in 22 patients with VGAM. The study aimed to gain insights into the effectiveness and safety of endovascular embolization as a treatment option for VGAM, as well as to evaluate the long-term outcomes and potential complications associated with this approach.

## 2. Materials and Methods

### 2.1. Study Design and Participants

This retrospective analysis focused on patients with VGAM who underwent one or more sessions of EE at the National Center for Neurosurgery in Astana, Kazakhstan, over a period of 14 years from January 2008 to December 2022. Initially, 31 patients who had undergone embolization were considered for the study. Among them, 71% of patients had valid contact details and were subsequently contacted for further assessment with MRI and angiograms, resulting in a final sample size of 22 patients who provided comprehensive and complete information for analysis. The patient cohort comprised newborns and children under the age of 18, all of whom were diagnosed with VGAM through MRI scans. The diagnosis was further confirmed using digital subtraction angiography (DSA) with a biplane angiographic system. The diagnostic images were thoroughly assessed by a multidisciplinary team of neuroradiologists, neurosurgeons, and pediatric specialists. The collected and analyzed data that included patients’ age, sex, clinical presentation, VGAM type based on Lasjaunias’ classification [8], the number of embolization sessions, fistula occlusion rate after EE, the need for additional surgery such as ventriculoperitoneal shunting (VPS) after EE, complications, and mortality. 

### 2.2. Endovascular Embolization Procedures

The criteria for considering endovascular embolization were evaluated based on the Bicêtre score [7], which assesses the status of the heart, lungs, central nervous system (CNS), kidneys, and liver functions (Table 1). The heart failure was assessed through the electrocardiogram and echocardiogram. The maximum score on this scale is 21 points. A score of ≤8 points indicated that surgical treatment carried a high risk of complications and mortality, even with EE. A score of 8–12 points indicated that although neurological outcomes might be favorable, urgent EE was required to stabilize the overall status due to refractory heart failure. A score of 13–21 points suggested postponing EE for up to 3–5 months, considering the higher risk of technical complications in the neonatal period due to smaller vessel caliber and unstable hemodynamic and somatic statuses.

The EE procedures were performed on all patients using the biplanar angiography suite. The approach for embolization involved a femoral transarterial approach. In most cases, newborns underwent EE after reaching five months of age, with only two cases receiving angiography and EE during the neonatal period. The procedures were performed using 4F–6F guiding catheters via the internal carotid artery (ICA) or vertebral artery (VA). After evaluating the angioarchitectonic anatomy of VGAM, an appropriate microcatheter or balloon catheter was positioned into the fistula to reduce blood flow. The embolization was carried out using microcoils and liquid embolizing agents.

### 2.3. Follow-Up

Postoperative follow-up and clinical examinations were assessed at 1 month after the procedures, followed by MRI scans at 3 months and angiography at 1 year or earlier if necessary. If multiple treatment sessions were required, subsequent sessions were scheduled at 3- to 6-month intervals.

In our study, we used a 5-point outcome score based on Jones et al.’s classification [13], with scores ranging from zero to four. A score of zero indicated death, while a score of four indicated normal neurological status. In our analysis, we simplified the scoring system by grouping patients into two categories. Scores between four and three were considered “good”, representing patients who had either normal neurological function or mild developmental delays. Scores two, one, and zero were categorized as “poor”, indicating patients with moderate developmental delays, severe developmental delays, or those who had passed away. This approach allowed us to effectively evaluate and compare the overall neurological and developmental outcomes of the patients in our study.

## 3. Results

Overall, 22 patients have been treated at the Vascular and Functional Neurosurgery Department, National Centre for Neurosurgery, since 2008. The average follow-up was 58.87 ± 48.11 months. The patients’ age ranged from 25 days to 17 years old, with a mean age of 37.85 months ± 58.96, with two neonates (9.1%) under one month, 17 infants (77.27%) aged between one month and two years, and three children (13.63%) of two years and older. Participants were predominantly male (86.36%). Three-fourths of participants had comorbidities, which included hydrocephalus, heart failure, disseminated intravascular coagulation syndrome, upper right segment bronchopneumonia, false croup, broncho-obstructive syndrome, chronic posterior cranial hematoma, and conditions after draining chronic hematoma. Most patients underwent EE using liquid embolizing agents, 11 (50%), and nearly 73% of patients experienced two and more sessions of intervention. Among them, three (13.63%) children required VPS due to persistent hydrocephalus even after EE (Table 2).

Based on VGAM classification, nine (40.9%) cases were classified as choroidal, and 13 (59.1%) cases were classified as mural. During the EE procedures, complete fistula occlusion was achieved in eight (36.3%) patients, while partial occlusion was achieved in 14 patients (63.7%). Postoperatively, two patients (9.1%) experienced tinnitus and diplopia due to midbrain and quadrigeminal ischemia.

The study reported a mortality rate of 9.1% (two cases): one patient died the day after surgery due to severe cardiac decompensation, and the other patient passed away after one month from pneumonia related to pulmonary hypertension, despite immediate improvement in cardiac function post-surgery.

In long-term assessments, four patients (18.2%) showed developmental delays, 16 patients (72.7%) had a positive outcome, and two of them have successfully graduated from secondary school.

### 3.1. Illustrative Cases: Case 1

A child aged 1.4 years presented with symptoms of food spitting and vomiting due to a VGAM, which was confirmed in brain ultrasonography and MRI findings. Subsequent cerebral angiography revealed the presence of multiple direct fistulae from the thalamoperforating artery, posterior cerebral artery, and posterior choroidal artery (PChA) to the vein of Galen (Figure 1).

To address the VGAM, the child underwent two embolization procedures via a transarterial approach. The second transarterial embolization resulted in complete occlusion of the lesion with no reported complications. Following the successful treatment, the child has been under dynamic observation. No further issues have been observed during the follow-up period (Figure 2).

### 3.2. Illustrative Cases: Case 2

At the age of 1.1 years, a young girl presented with symptoms of unconsciousness and an increase in head size since birth. After MRI, she was diagnosed with a VGAM. Further cerebral angiography revealed the presence of multiple direct connections between arteries, including the thalamoperforating artery, posterior cerebral artery, posterior choroidal artery, and pericallosal arteries, to the vein of Galen (Figure 3).

To treat the VGAM, the child underwent two embolization procedures through transarterial embolization. Over the course of two sessions, the lesion was completely occluded without any reported complications. Following the successful treatment, the child has been under observation, and no issues have been observed during the follow-up period (Figure 4).

## 4. Discussion

Since VGAM was first described in 1895 by Steinheil, this malformation has been known by various names, such as “varix aneurysm”, “aneurysms of the vein of Galen”, “vein of Galen aneurysmal malformations”, and “arteriovenous aneurysms of the vein of Galen” [14]. Despite the different terms used, the common characteristic among all cases is the dilatation of the Galen vein at various stages.

Over the years, the medical community has been researching different interventions and surgical treatment methods for VGAM. One of the significant techniques proposed by Boldrey and Miller in 1949 involves the posterior cerebral artery and its arterial feeders clipping [15]. Additionally, transarterial embolization, introduced by Lasjaunias in 1987, and access via the femoral vein and torcular sinus, described by Mickle in 1986 and Dowd in 1990, have been studied as treatment options for this malformation [16,17,18]. 

Currently, endovascular therapy remains the preferred first-line treatment for VGAM, with the lowest mortality and complication rates and favorable clinical outcomes. 

Based on recent research findings, the optimal timing for the initial endovascular intervention is recommended at approximately 5–6 months after birth [19]. However, in certain cases, emergency endovascular treatment may be necessary, particularly for reducing shunting in neonates suffering from congestive cardiac failure that do not respond well to medical therapy [20]. 

In recent years, the key perspective in diagnostics has shifted to the significance of prenatal diagnosis of VGAM, which holds immense importance in providing early intervention with subsequent appropriate management. Ultrasound and MRI have emerged as crucial complementary diagnostic tools for detecting VGAM during pregnancy. These imaging techniques not only contribute to an accurate diagnosis but also play a vital role in prognostic evaluation. Furthermore, by incorporating genetic testing into the diagnostic process, healthcare professionals can identify specific genetic factors or mutations associated with VGAM in the early stages of pregnancy planning. Thereby, the integration of genetic testing and personalized medicine in the management of VGAM holds great promise for improved diagnostic accuracy, targeted treatment approaches, and better outcomes for patients and their families [5]. 

Clinical presentations and angioarchitectures of VGAMs are known to be related to the age at presentation, and the main objective of investigators is to provide the best care and ensure successful treatment for individuals affected by the disease [21]. A meta-analysis conducted by Jun Yan et al. revealed that the mortality rate without treatment reaches 47%, whereas endovascular intervention reduces it to 12% [12]. Considering patients’ somatic status is crucial in the improvement of treatment outcomes. Another meta-analysis by W. Brinjikji et al. showed that patients treated by endovascular intervention had a higher favorable neurological outcome in infants and older children (77%) compared to newborns (48%). Additionally, the neonatal group experienced higher perioperative risks, including hemorrhage, ischemia, adverse outcomes, and increased mortality rates (27%). In contrast, older children had a much lower mortality rate of about 1% [2]. These findings are in line with our results and suggest that the success of endovascular intervention may vary depending on the age group, with better outcomes observed in infants and older children compared to newborns. Both of our fatal cases occurred in infants with low Bicêtre scale scores.

Timely implementation of endovascular therapy significantly increases the chances of a good outcome in patients with VGAM. Choosing the “optimal intervention period” is a key factor that affects complication risks and treatment outcomes. The use of the Bicêtre scale can improve treatment outcomes and aid clinicians in deciding the “optimal intervention period”, considering the patient’s age and medical status [22]. By considering these factors, healthcare professionals can enhance the effectiveness of interventions and improve the overall prognosis for patients with VGAM.

Treatment for VGAM involves a range of approaches to address the specific manifestations of the condition. The clinical symptoms of VGAM are distinct for each age group [23]. Neonates typically exhibit severe cardiac failure, with high-flow shunts being predominant in this group. Infants commonly present with symptoms like increased head circumference, hydrocephalus, or seizures, while older children tend to experience milder symptoms, such as headaches. Our study has also found that clinical signs differ depending on patients’ age group. These distinct clinical manifestations at different ages highlight the importance of considering age as a crucial factor in diagnosing and managing VGM cases.

Such clinical manifestations of VGAM, including cardiopulmonary insufficiency and pulmonary hypertension, mostly occur in patients during the neonatal period. To address these signs, conservative therapy, such as diuretics, are managed through inotropic therapy and anticonvulsants [24]. 

The etiology of hydrocephalus associated with VGAM is multifactorial, involving two main factors: Sylvian aqueduct obstruction and impaired cerebrospinal fluid resorption, also known as resorptive block [25,26]. Due to the multifactorial nature of this condition, the treatment of VGAM-associated hydrocephalus can vary. In some cases, resolution can be achieved through surgical interventions such as VPS or endoscopic triventriculostomy surgeries [27,28]. Spontaneous thrombosis of VGAM has also been reported in the literature after VPS operations [29].

In our center, we prioritize the endovascular approach as the first step of treatment due to its promising results in achieving hydrocephalus resolution. The literature also supports the effectiveness of endovascular therapy in this regard [30], which is confirmed by our study’s findings. Therefore, ventriculoperitoneal shunting should be considered only for patients who do not experience improvement after embolization or for those who have hydrocephalus not related to VGAM.

The choice of treatment should be tailored to each patient’s specific characteristics and clinical presentations, taking into consideration the advantages and potential risks of different approaches. 

The use of staged sessions during EE is a common practice at our center to effectively reduce complications. This approach is supported by Jun Yan et al. [12], who clearly demonstrated a lower complication rate compared to the one-step procedures. The next specific steps should be taken include appropriate superselective catheterization and embolization, preserving key vessels like thalamoperforants and choroidal branches, and selecting an appropriate embolizing agent while maintaining normal deep vein outflow. Furthermore, the presence of normal drainage between the deep venous system and VGAM should be considered before embolization.

Transarterial embolization is a specialized procedure employed to manage high-flow fistulae. In this method, a dedicated microcatheter is carefully guided by the natural blood flow. Coils or N-butyl-cyanoacrylate (n-BCA) are recommended for an application aimed at intentionally decreasing blood flow in these fistulae. After a reduction in fistulae flow achieved by the prior use of coils or n-BCA, the introduction of liquid embolizing agents is proposed. In particular, these liquid agents offer distinct advantages when the process requires sustainable recruitment management [31].

Overall, careful consideration of the specific type of VGAM and employing the appropriate treatment techniques can lead to outcomes that are more successful and reduce complications for affected patients.

## 5. Conclusions

In conclusion, the primary objective in treating VGAM is to stabilize cardiac and systemic complications until endovascular intervention can be performed. Given the complexity of VGAM, a multidisciplinary approach is recommended to optimize treatment outcomes. Combining endovascular therapy with a comprehensive management strategy has been shown to reduce mortality rates and may lead to normal neurological development in patients.

It is important to note that the main goal of EE is not necessarily complete fistula occlusion, but rather to eliminate or reduce the arteriovenous shunt, leading to improved cardiac output and neurological development. The focus of treatment is on restoring normal neurocognitive and clinical parameters, rather than solely relying on the angiographic picture.

## Figures and Tables

**Figure 1 diagnostics-13-02704-f001:**
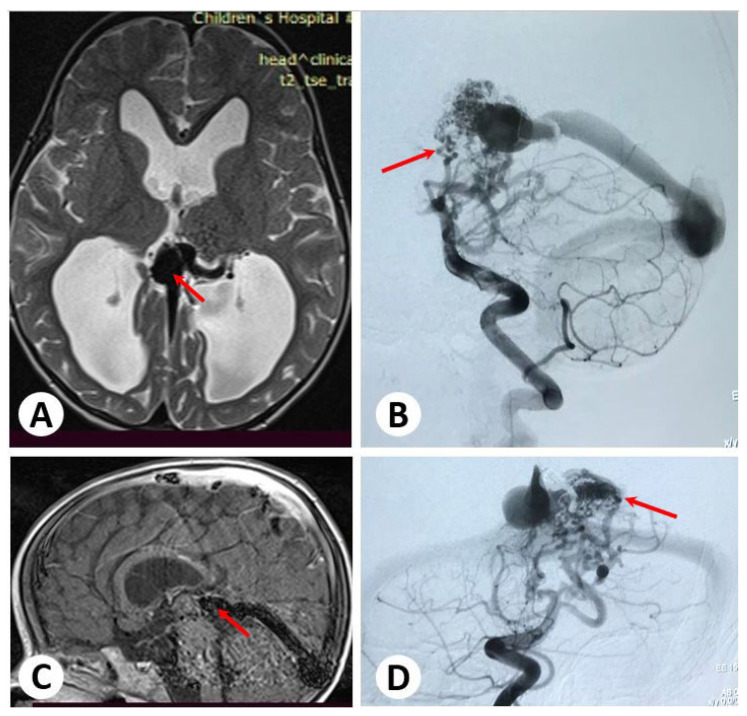
Brain magnetic resonance axial and sagittal images showing a dilated vein of Galen and enlarged ventricles (**A**,**C**). Cerebral angiography (**B**,**D**) reveals multiple direct fistulae from the thalamoperforating artery and posterior cerebral artery to the vein of Galen.

**Figure 2 diagnostics-13-02704-f002:**
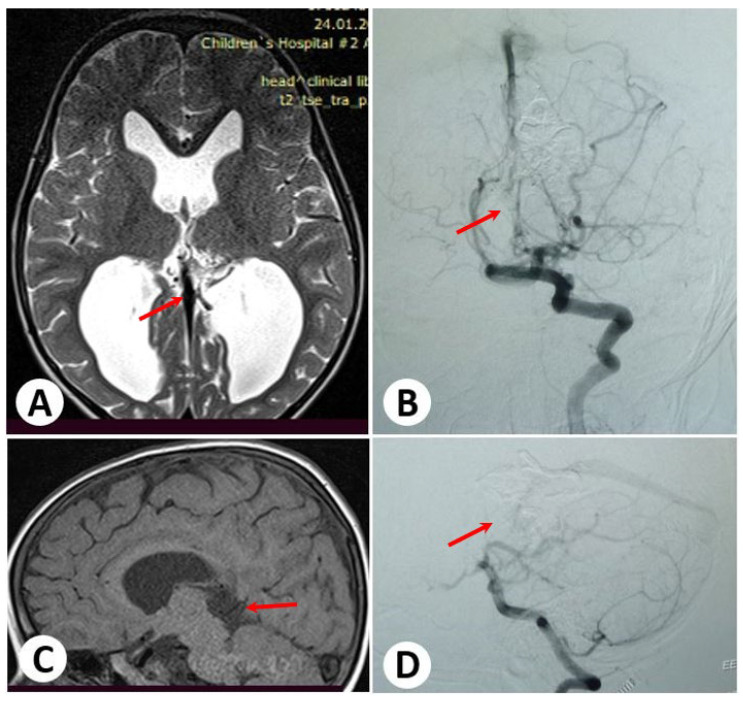
Follow-up images after 12 months. Brain magnetic resonance axial and sagittal imaging (**A**,**C**) indicates a reduction in the size of the vein of Galen and ventricles. Cerebral angiography (**B**,**D**) shows subtotal occlusion of the lesion. There is no evidence of any remaining issues.

**Figure 3 diagnostics-13-02704-f003:**
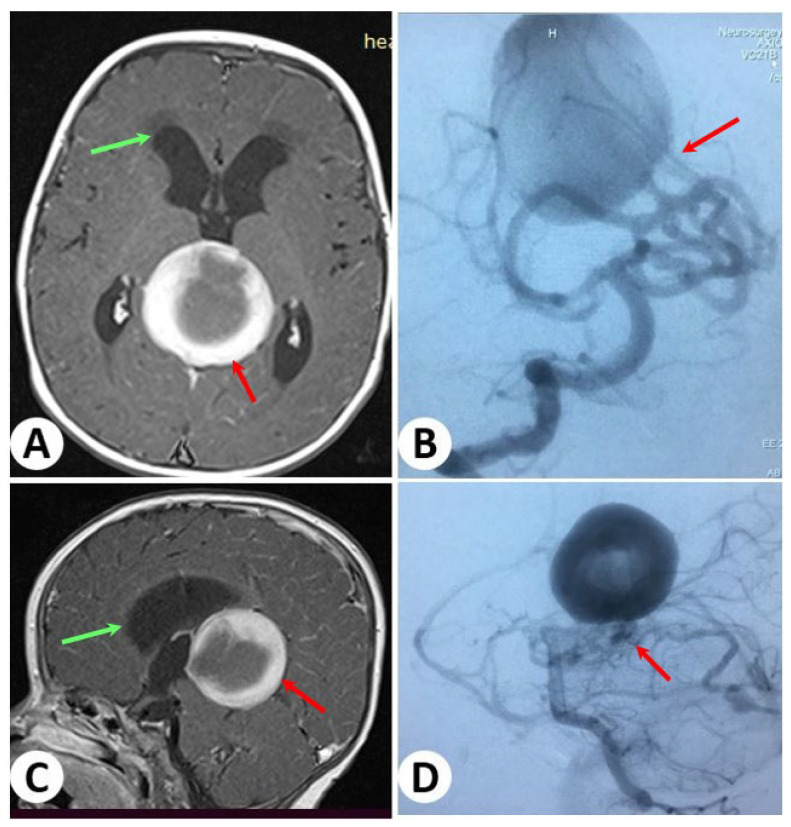
Brain magnetic resonance axial and sagittal images showing a dilated vein of Galen (red arrows) and enlarged ventricles and periventricular edema (green arrows) (**A**,**C**). Cerebral angiography (**B**,**D**) reveals multiple direct fistulae from the thalamoperforating artery and posterior cerebral artery to the vein of Galen.

**Figure 4 diagnostics-13-02704-f004:**
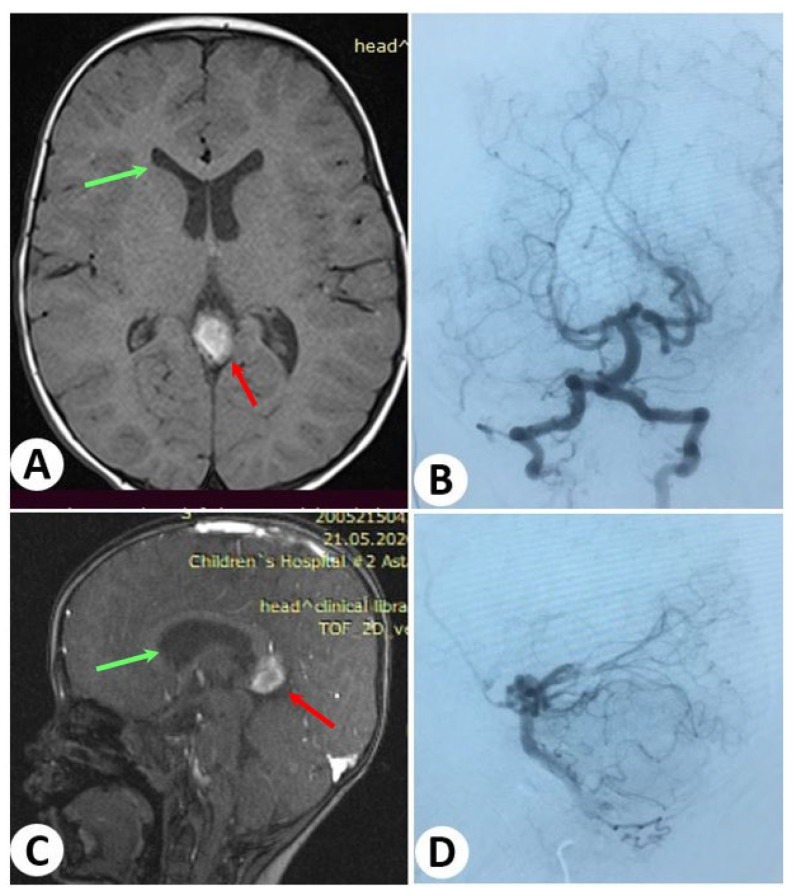
Follow-up images after 18 months. Brain magnetic resonance axial and sagittal imaging (**A**,**C**) revealed a significant reduction in the size of the vein of Galen (red arrows) and ventricles, with the periventricular edema resolved (green arrows). Additionally, cerebral angiography (**B**,**D**) demonstrated the complete occlusion of the lesion. There is no evidence of any remaining issues.

**Table 1 diagnostics-13-02704-t001:** Bicêtre score.

Points	Cardiac Function	Cerebral Function	Respiratory Function	Hepatic Function	Renal Function
5	Normal	Normal	Normal	-	-
4	Overload, no medical treatment	Subclinical, isolated EEG ^a^ abnormalities	Tachypnea finishes bottle	-	-
3	Failure, stable with medical treatment	Nonconvulsive intermittent neurologic signs	Tachypnea does not finish bottle	No hepatomegaly, normal hepatic function	Normal
2	Failure, not stable with medical treatment	Isolated convulsion	Assisted ventilation, normal saturation FiO2 ^b^ < 25%	Hepatomegaly, normal hepatic function	Transient anuria
1	Ventilation necessary	Seizures	Assisted ventilation. Normal saturation FiO2 > 25%	Moderate or transient hepatic insufficiency	Unstable diuresis with treatment
0	Resistant to medical therapy	Permanent neurological signs	Assisted ventilation, desaturation	Abnormal coagulation, elevated enzymes	Anuria

^a^ EEG: Electroencephalogram, **^b^** FiO2: Fractional inspired oxygen; Maximal score: 5 (cardiac) + 5 (cerebral) + 5 (respiratory) + 3 (hepatic) + 3 (renal) = 21 [7].

**Table 2 diagnostics-13-02704-t002:** Demographic and clinical characteristics of the participants.

Patients	*N* = 22, (100%)
Age, mean ± SD	37.85 months ± 58.96
neonates (<1 month)	2 (9.1%)
infant (≥1 month to <2 years)	17 (77.27%)
child/adult (≥2 years)	3 (13.63%)
Sex	
female	3 (13.63%)
male	19 (86.36%)
Comorbidities	
hydrocephalus	12 (54.54%)
heart failure	5 (22.73%)
pulmonary hypertension	2 (9.1%)
hemorrhage	3 (13.63%)
Number of endovascular sessions	
1	6 (27.27%)
2	9 (40.91%)
3 and more	7 (31.83%)
Occlusion Rate	
total	8 (36.3%)
partial	14 (63.7%)
Type of intervention	
liquid embolizing agents	11 (50%)
coil	4 (18.18)
combination of glue and coil	7 (31.82%)
ventriculoperitoneal shunting	3 (13.63%)
VGAM Type	
choroidal type	9 (40.9%)
mural type	13 (59.1%)
Long-Term Outcomes	
good outcome	16 (72.7%)
poor outcome	6 (27.3%)
Mortality	2 (9.1%)

## Data Availability

The data presented in this study are available on request from the corresponding author. The data are not publicly available due to ethical issues.

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
