# Peer review of "Long-Term Outcomes of Endovascular Embolization in a Vein of Galen Aneurysmal Malformation: A Single-Center Experience"

_diagnostics, 2023, doi:10.3390/diagnostics13162704_

Round 1
Reviewer 1 Report
Thank you for the opportunity to review this interesting article. The work is generally well written. However, the following issues require clarification: How was heart failure assessed in the presented group? Has the level of NTproBNP been assessed? Was elevated NTproBNP associated with poorer prognosis? Thank youAuthor Response
Dear Reviewer,
I would like to extend my heartfelt appreciation for dedicating your time and expertise to review our manuscript titled "Long-Term Outcomes of Endovascular Embolization in Vein of Galen Aneurysmal Malformation: Single-Center Experience".
Below, I have provided a point-by-point response to your feedback:
Point 1: How was heart failure assessed in the presented group?
Response 1: Heart failure was assessed through the electrocardiogram and echocardiogram. The relevant text was added in the section “2.2 Endovascular embolization procedures”. Page 3, lines 103-104.
Point 2: Has the level of NTproBNP been assessed?
Response 2: The data on the level of NTproBNP has not been assessed due to the lack of information available. Presurgery analyses were performed in the hospital for Maternal and child health.
Point 3: Was elevated NTproBNP associated with poorer prognosis?
Response 3: Information on the NTproBNP analysis was not available in our Centre, so we could not assess the association of elevated NTproBNP with poorer prognosis in our study.
Reviewer 2 Report
In this study, authors reported the long-term outcomes of patients with Vein of Galen aneurysmal malformation (VGAM), who were treated with endovascular embolization, via focusing on VGAM patients who underwent one or more endovascular embolization sessions between January 2008 and December 2022. Data encompassed clinical and demographic characteristics, types of endovascular embolization, treatment complications, mortality rates, and long-term outcomes. This study showed that combining endovascular therapy with a comprehensive management strategy can significantly reduce mortality rates and improve the possibility of normal neurological development in patients. This is a Single-Center Experience retrospective analysis, authors are advised to solve the above issues.
1. Minor typos: please add a blank before and after “+” “=” “±” et al. Figure 1.a should be Figure 1a. The image labels of all figures (a: ABCD, b: ABCD) are very confusing.
2. In the introduction section, authors presented a comprehensive introduce of the background of VGAM, as a rare congenital cerebral vascular condition with a high mortality rate, its relationship with specific genes remains unknown. However, authors are failed in giving a comprehensive review of the current progress of endovascular embolization (EE) treatment.
3. In section 3, authors declared the Illustrative cases in section 3.1, where is section 3.2?
4. Especially, in section 2.2 “Endovascular embolization procedure (actually, “procedure” should be “procedures”)”, authors only pointed out that the “The embolization was carried out using microcoils and liquid embolizing agents (PHIL, Onyx, MagicGlu).” Whether other embolic agents can be used clinically?
5. In PHIL, Onyx, MagicGlu, which one performs the better or best? Should be discussed.
6. How about the development prospect of endovascular embolizing agents? As the experts in this field, authors are encouraged to give their comments. A paper on the design of novel embolic agents must to be cited: https://doi.org/10.1016/j.msec.2022.112677.
Author Response
Dear Reviewer,
I want to express my sincere gratitude for investing your time and expertise in evaluating our manuscript. Your efforts in reviewing our work are greatly appreciated.
Your thoughtful comments and suggestions have been invaluable in enhancing the quality and depth of our work.
We have carefully considered each of your comments and have made significant revisions to address the concerns you raised. Below, I have provided a point-by-point response to your feedback:
Point 1: Minor typos: please add a blank before and after “+” “=” “±” et al. Figure 1.a should be Figure 1a. The image labels of all figures (a: ABCD, b: ABCD) are very confusing.
Response 1: the blanks before and after “+” “=” “±” et al. were added through the text. The image labels were changed to be less confusing. Now there are four figures with ABCD under labels.
Point 2: In the introduction section, authors presented a comprehensive introduce of the background of VGAM, as a rare congenital cerebral vascular condition with a high mortality rate, its relationship with specific genes remains unknown. However, authors are failed in giving a comprehensive review of the current progress of endovascular embolization (EE) treatment.
Response 2: A review of the current progress of endovascular embolization was added in the introduction. Page 2, lines 55-75.
Point 3: In section 3, authors declared the Illustrative cases in section 3.1, where is section 3.2?
Response 3: The section 3.2 were added.
Point 4: Especially, in section 2.2 “Endovascular embolization procedure (actually, “procedure” should be “procedures”)”, authors only pointed out that the “The embolization was carried out using microcoils and liquid embolizing agents (PHIL, Onyx, MagicGlu).” Whether other embolic agents can be used clinically?
Response 4: “Procedure” was replaced with “procedures” throughout the text. We have listed the embolic agents available in our hospital. Other embolic agents can also be used clinically. The names of embolic agents have been removed from the text, so as to prevent the possibility of favoring any specific embolic agent.
Point 5: In PHIL, Onyx, MagicGlu, which one performs the better or best? Should be discussed.
Response 5: Our intention is to avoid highlighting the significance of any specific agent, as this could potentially lead to conflicts of interest. The names of embolic agents have been deliberately excluded from the text. Our study does not seek to make comparisons between these agents; rather, its focus is on exploring the clinical outcomes of endovascular embolization, irrespective of the specific embolic agent used. However, a discussion of applications of embolic agents was added in the discussion section, page 9, lines 298-304.
Point 6: How about the development prospect of endovascular embolizing agents? As the experts in this field, authors are encouraged to give their comments. A paper on the design of novel embolic agents must to be cited: https://doi.org/10.1016/j.msec.2022.112677.
Response 6: the citation with appropriate text has been added. Page 2, lines 64-69.